# Divergence of Chemerin Reduction by an ATS9R Nanoparticle Targeting Adipose Tissue In Vitro vs. In Vivo in the Rat

**DOI:** 10.3390/biomedicines10071635

**Published:** 2022-07-07

**Authors:** Alexis Orr, Kunli Liu, Adam E. Mullick, Xuefei Huang, Stephanie W. Watts

**Affiliations:** 1Department of Pharmacology and Toxicology, Michigan State University, East Lansing, MI 48824, USA; orralexi@msu.edu; 2Department of Chemistry, Michigan State University, East Lansing, MI 48824, USA; liukunli@chemistry.msu.edu; 3IONIS Pharmaceuticals, Carlsbad, CA 92010, USA; amullick@ionisph.com; 4Departments of Chemistry and Biomedical Engineering, Michigan State University, East Lansing, MI 48824, USA; xuefei@chemistry.msu.edu

**Keywords:** chemerin, adipose tissue, nanoparticle, ATS9R, ASO

## Abstract

Nanoparticles (NPs) can enable delivery of a drug to a targeted tissue. Previous studies have shown that an NP utilizing an adipose targeting sequence (ATS) peptide in conjunction with a drug can selectively deliver the drug to mouse adipose tissues, using the prohibitin protein expressed in adipose tissue as the target of the ATS. Adipose tissue is a major source of the adipokine chemerin, a prohypertensive protein. Liver-derived chemerin, the largest source of circulating chemerin, is biologically inactive in blood pressure regulation. Our goal is to understand if chemerin produced in adipose tissue contributes to blood pressure/hypertension. We hypothesize the ATS drug delivery system could be used specifically to reduce the levels of adipose tissue-derived chemerin. We created an NP consisting of an antisense oligonucleotide (ASO) against chemerin and a FITC-labeled ATS with a nine arginine sequence (ATS9R). In vitro studies showed that the ASO is functional when incorporated into an NP with ATS9R as it reduced chemerin mRNA expression in isolated epidydimal (Epi) and retroperitoneal (RP) fat adipocytes from Dahl SS rats. This same NP reduced chemerin in isolated whole fats. However, this NP was unable to selectively deliver the ASO to adipose tissue in vivo; liver delivery was dominant. Varying NP doses, administration route, and the concentration of components constituting the NP showed no improvement in ASO delivery to fats vs. the liver. Further studies are therefore needed to develop the ATS9R system to deliver an ASO to adipose beds in rats.

## 1. Introduction

Obesity has become one of the largest threats to human health, specifically with its often-seen comorbidity of hypertension [1,2,3,4]. Due to the excessive fat burden seen in obese conditions, fat-derived substances (adipokines) must be considered in the effort to understand the pathophysiology of obesity-associated hypertension. Identifying such substances would aid physicians in their fight against obesity-associated hypertension, an area of medicine that is significantly lacking in effective therapeutic interventions. While there are numerous substances derived from fat that could be targets for this type of therapy, the adipokine chemerin is a leading contender. There is a strong positive association between circulating chemerin levels and both systolic and diastolic blood pressure, arterial stiffness, body mass index (BMI), and visceral fat burden, observed in both human and animal populations [4,5,6,7,8,9,10,11,12,13,14,15,16,17,18,19,20,21,22,23,24,25].

Translated from the gene *Rarres2*, chemerin is secreted primarily by the liver [19,20,26]. Our laboratory has discovered that while most circulating chemerin is produced by the liver in the rat, liver-derived chemerin is biologically inactive relative to blood pressure regulation. However, whole-body reduction of chemerin lowers not only normal blood pressure, but also the hypertension stimulated by a high fat diet in the Dahl SS rat [27]. These findings were based on the use of a whole-body antisense oligonucleotide (ASO) and an ASO made specifically for the liver [28,29]. If the liver was not responsible for the chemerin that supported blood pressure, the logical conclusion is that the adipose tissue, the second greatest contributor of chemerin, produces the biologically important chemerin.

Studies have shown some promise in creating a system that allows for adipose tissue targeted drug delivery through the use of nanoparticles (NPs). The group of Kolonin et al. [30] utilized an in vivo phage display to identify a peptide motif (CKGGRAKDC) that interacts with the protein prohibitin [31]. Prohibitin, while cytosolic in many cell types, is also localized to the plasma membrane of adipocytes and endothelial cells of adipose vasculature [32,33]. The interaction of this peptide motif (considered the adipose targeting sequence (ATS)) with prohibitin allows for uptake of substances through receptor-mediated endocytosis [34]. The cell penetrating ability of the ATS sequence was later improved by adding a 9-arginine (R) sequence [32]. This improved ATS9R complex showed robust uptake by prohibitin in mouse adipocytes, delivering a short hairpin RNA for the fatty acid binding protein 4, all while avoiding liver uptake. Other publications support the use of an ATS/ATS9R NP in mice [32,35,36].

We hypothesize that an NP utilizing the above described ATS9R peptide can be used to reduce the levels of adipose tissue-associated chemerin. We chose to focus our efforts on the Dahl SS rat. This strain of rat—both male and female—develop a robust hypertension from high fat feeding [10,37]. We tested whether an FITC-labeled ATS9R NP carrying an ASO against chemerin would reduce chemerin expression in an adipose-selective manner. This ATS9R NP carrying the ASO was used in complementary in vitro, ex vivo, and in vivo experiments. While excellent knockdown of chemerin was observed in vitro in adipose tissues, our results do not support the adipose tissue specificity of an ATS9R-based NP in vivo in the rat.

## 2. Materials and Methods

### 2.1. Animal Use

Normal male 200–250 g Dahl SS rats (Charles River, Indianapolis, IN, USA) were used following the Guide for the Care and Use of Laboratory Animals (8th edition, 2011). Pentobarbital (80 mg/kg intraperitoneal) was used to anesthetize rats and a bilateral pneumothorax was created for dissection. The following tissues were removed for measurements: thoracic aorta, subscapular brown fat, white epidydimal (Epi) fat, white retroperitoneal (RP) fat, fat located on top of the proximal end of the mesenteric arcade (mes fat), liver, duodenum, skeletal muscle (hind leg region), bladder, adrenals, heart, kidney, lung, spleen, stomach fundus, and trachea. Adipose tissue immediately surrounding an artery was considered its respective PVAT. Using a stereomicroscope and microscissors, PVAT was removed from vessels (APVAT from aorta) in Krebs−Ringer bicarbonate buffer (KRBB; 135 mmol/L NaCl, 5 mmol/L KCl, 1 mmol/L K_2_HPO_4_, 5.5 mmol/L glucose, 20 mmol/L HEPES, 10 mL antibiotic/antimycotic, pH to 7.4). The Michigan State University Institutional Animal Care and Use Committee (IACUC) guidelines (Protocol # 02-18-026, approved 18 February 2018) were followed for all protocols.

### 2.2. Adipocyte and SVF Isolation

Dissected Epi fat and RP fat were placed in sterile 1.5 mL tubes with 1 mg of collagenase type I (LS004196, Worthington Biochemical, Lakewood, NJ, USA) and 4% BSA (A2153, Sigma, Burlington, MA, USA) in KRBB. Adipose tissue was minced with small scissors and tubes placed in a rotating incubator at 37 °C for 45 min, or until the solution was opaque with no large adipose pieces visible. After digestion, tubes were centrifuged at 200× *g* for 5 min at room temperature, subnatant was discarded and the stromal vascular fraction (SVF) pellet was collected using a sterile 23 gauge needle. Eight hundred (800) µL of KRBB was added to each tube of adipocytes and tubes were gently mixed by finger flicking. Tubes were again centrifuged at 200× *g* for 5 min and the subnatant discarded, for a total of 3 washes. Adipocytes were collected for mRNA isolation.

Following SVF isolation from adipocytes, the SVF pellet was centrifuged in a new sterile 1.5 mL tube at 1500× *g* for 5 min. The supernatant was removed and 800 µL of KRBB was added, for a total of 3 washes. The SVF pellet was collected for mRNA isolation.

### 2.3. NP Formation

ATS9R was custom synthesized by Glbiochem (Shanghai, China) and tagged with FITC. The structure of the peptide was confirmed with high resolution mass spectrophotometry. The ASO was synthesized by IONIS Pharmaceuticals (Carlsbad, CA, USA) with the following sequence: ASO Gen 2.5 (5′-3′): GTTTTATTAGCCTGGA.

The NP complex was synthesized using electrostatic complexation following a similar approach previously reported [32,38,39,40]. Five hundred (500) µL of 10 mg/mL ATS9R solution was added drop-wise to 500 µL of a 10 mg/mL ASO solution under vigorous stirring. Once all solutions were combined, the mixture was allowed to stir for 15 min at room temperature. The mixture was then centrifuged at 1500× *g* for 5 min to remove free ATS9R and ASO. After centrifugation, the pellet was collected and redispersed into 500 µL of water and the pH was adjusted to 7.0 using 0.5 mol/L NaOH to create the 1:1 NP formulation.

To create the NP formulation with the ratio of 0.25:1 of ATS9R:ASO, 2.5 mg/mL ATS9R was used following the procedure above.

### 2.4. Characterization of NP Complex

NP size was measured by dynamic light scattering (DLS) using a Zetasizer Nano apparatus (Malvern, UK). The NP was imaged under transmission electron microscope (TEM) (JEM-2200FM, Jeol Ltd., Akishima, Tokyo, Japan) operating at 200 kV using a Gatan multiscan CCD camera with Digital Micrograph imaging software (Gatan, Pleasanton, CA, USA). Uranyl acetate staining was employed for TEM imaging. UV spectrophotometry was used to quantify the concentration of ATS9R and ASO within the NP complex. Individual standard curves of ATS9R and ASO in solution were created to quantify samples with an absorbance at 260 nm indicative of the ASO and that at 450 nm indicative of the ATS9R peptide. Using these values, the absorbance values at 260 nm and 450 nm of the NP complex could be used to calculate the relative ratio of ASO and ATS9R in the NP based on the respective standard curves based on the Beer−Lambert law. When a 1:1 ASO and ATS9R peptide were mixed together, the molar ratio of ASO and ATS9R in the resulting NP was determined to be 1:1 from the absorbance values.

### 2.5. RT-PCR

Tissue samples were homogenized using the Omni Bead Ruptor (Omni, Inc., Jennesaw, GA, USA) and the RNA Lysis Buffer (R1060, Zymo Research, Irving, CA, USA), except for skeletal muscle and bladder where RNA RLT (cat # 1015750, Qiagen, Valencia, CA, USA) was used. RNA from all tissues except skeletal muscle and bladder was isolated using the Zymo Quick-RNA Mini Prep Kit (cat # R1055, Zymo Research, Irving, CA, USA). Skeletal muscle and bladder RNA was isolated using RNeasy Fibrous Tissue Mini Kit (cat # 74704, Qiagen, RRID: SCR_008539). All isolated RNA was quantified using a Nanodrop 2000c spectrophotometer (Thermo Fischer Scientific, Waltham, MA, USA). The High-Capacity cDNA Reverse Transcription Kit (4368814, Thermo Fischer Scientific) was used to reverse transcribe all cDNA. RT-PCR was performed using PerfeCTa FastMix II Rox (cat # 95119-012, QuantaBio, Beverly, MA, USA) on an Applied Biosystems 7500 FAST Real-Time PCR System (Thermo Fischer Scientific, RRID: SCR_018051, Waltham, MA, USA) using cycle parameters of 10 min at 95 °C, 15 s at 95 °C, and 1 min at 60 °C for 40 cycles, followed by a melt curve to determine a single PCR product. Measures were compared by running the housekeeping gene *β-actin* (ACTB). Primers were obtained from Integrated DNA Technologies (IDT; Coralville, IA, USA) and are as follows:

*rRarres2* (chemerin) (306118822)

Primer 1: GAGCTTAAATTCCAGCCTCACAA

Primer 2: CAGGAGATCGGTGTGGACAGT

Probe: /56-FAM/TGATGACCTGTTCTTCTCAGCTGGCACC/36-TAMSp/

*β-actin* (306118826)

Primer 1: TCACTATCGGCAATGAGCG

Primer 2: GGCATAGAGGTCTTTACGGATG

Probe: /56-FAM/TCCTGGGTATGGAATCCTGTGGC/36-TAMSp/

### 2.6. In Vitro NP Incubation

Adipocytes were isolated from Epi and RP fat as described above. Fifteen (15) µL of adipocytes were incubated with 30 µL of acridine orange/propidium iodide (AOPI) stain (cat **#** CS2-0103, Nexcelom Biosciences, Lawrence, MA, USA) and 15 µL of KRBB for 1 min and then counted on a Cellometer Vision (Nexcelom Bioscience). Live cell concentration was determined and diluted to 2.5 × 10^6^ cells/mL. Live adipocyte percentages above 60% of the total population were considered acceptable. Five hundred (500) µL of adipocytes were then added to 500 µL of DMEM/F12 adipocyte maintenance media with 10% FBS in a sterile 1.5 mL tube, containing a final concentration of either 54 µg/mL of NP (determined by ASO concentration), 54 µg/mL of ASO, or nothing as a vehicle control. Cells were left in the media for 24 h at 37 °C and washed three times the following day with 500 µL of adipocyte media. Cells were then collected for RNA isolation.

### 2.7. Ex Vivo NP Incubation

Epi and RP fat were removed from a non-treated animal and cut into 100 mg of intact fat pieces. Pieces were placed in a 48-well plate with 500 µL of adipocyte media containing a final concentration of either 0.002 mg/mL NP (determined by ASO concentration), 0.002 mg/mL ASO, or nothing as a vehicle control. The plate was covered and placed in a 37 °C incubator for 24 h. The following day, the tissues were washed ten times with adipocyte media and collected for RNA isolation.

For experiments observing the quenching effect within the NP, 100 mg of intact fat pieces of Epi and RP fat from an untreated animal were incubated for 24 h at 37 °C in 500 µL of adipocyte media containing a final concentration of either 0.002 mg/mL NP (calculated using ASO concentrations), 0.002 mg/mL ATS9R, or a vehicle control. The following day, tissues were washed ten times with adipocyte media and imaged on the IVIS Spectrum In Vivo Imaging System (IVIS; PerkinElmer Inc., Waltham, MA, USA, RRID: SCR_018621).

### 2.8. In Vivo NP Incubation

To determine the distribution of the NP and ability of the NP to deliver an ASO against chemerin, animals were anesthetized using 1–2% isoflurane, weighed, and injected with the NP, either subcutaneously or intravenously (tail vein). Doses were determined using ASO concentration.

#### 2.8.1. Subcutaneous Injections

Rats were injected with a vehicle control (phosphate-buffered saline, PBS), 10 or 25 mg/kg of ASO/NP, both with the 1:1 and 0.25:1 ratios of the ATS9R molecule to the ASO molecule. Two days following the injection, the animals were sacrificed under 1–2% isoflurane and pneumothorax, and samples were collected. Parts of the liver, Epi and RP fat were flash frozen in liquid nitrogen for RNA isolation. Liver, Epi fat, RP fat, mes fat, heart, lung, thoracic aorta + PVAT, kidney, spleen, and adrenals were removed and weights were recorded, placed in tubes with PBS, and kept on ice for imaging on the IVIS (2.10).

#### 2.8.2. Tail Vein Injections

Rats were anesthetized using 1–2% isoflurane and weighed. A catheter was created using a blunt tip 25-gauge needle, P10 tubing, and the tip of a 25-gauge ½ inch needle. The catheter was inserted into the tail vein (lateral on tail) and flushed with sterile normal saline. The dose of NP/ASO was injected and followed by another normal saline flush. Two days following injection, the same euthanasia and tissue collection procedure was followed as described above. For vehicle-treated tail vein injections, normal saline was used in place of a vehicle (PBS) to reduce the risk of harm to the animals.

### 2.9. IVIS Imaging

After tissues were removed from the animal and placed in cold PBS, tissues were immediately taken to the IVIS for imaging. Tissues were removed from cold PBS and placed on black ArtAgain paper as a standard black background for imaging. The paper containing tissue specimens was then placed on a black plastic sheet and placed into the IVIS. Tissues were imaged and radiance for each tissue was recorded (with regions of interest), and then normalized to radiance per gram of tissue weight. Imaging occurred within 1 h of tissue removal from the animal.

### 2.10. Data Analysis

All values reported represent means ± SEM for a number of biological replicates (*n*). Radiance measures for images taken on the IVIS were determined by Living Image software (version 4.5.2, PerkinElmer Inc, Waltham, MA, USA, RRID: 014247). Images taken on the IVIS used the same exposure and scale, set equally for all tissues in experimental sets. For those groups with three or greater biological replicates, a student’s *t*-test was used when comparing two groups and a one-way ANOVA with a Tukey correction when comparing more than two groups. A *p* < 0.05 was considered statistically significant. All data were graphed using Graph Pad Prism (GraphPad Software, San Diego, CA, USA, RRID: SCR_002798).

## 3. Results

### 3.1. Liver Expressed the Most Chemerin mRNA, Followed by Adipose Tissue

RT-PCR was used to determine that, in the male Dahl SS rat, the liver expressed the highest amount of chemerin mRNA compared to the other samples studied (Figure 1). Samples with the next highest expression of basal chemerin mRNA included adipose tissues such as aortic PVAT (APVAT), brown subscapular fat (brown fat), epidydimal (Epi), and retroperitoneal (RP) fats. Adipose tissues are thus a logical target for ASO delivery.

As adipose tissue contains several different types of cells, we conducted further analyses to understand what cell type was the primary contributor of chemerin mRNA in adipose tissue. Within the adipose tissues of Epi fat and RP fat, adipocytes expressed chemerin mRNA at significantly higher levels than the SVF (Figure 2). This gave us the confidence to move forward with creating an NP to selectively target adipocytes.

### 3.2. NP Was Synthesized Reproducibly and Was Stable

The NP was synthesized through the complexation of the ATS9R peptide and ASO (Figure 3A). As shown in pictures in Figure 3B, upon mixing the ATS9R and ASO, particles were formed which could be spun down to remove any unbound components. The pellet could be redispersed back into the water. In contrast, the ATS9R peptide or ASO alone in water did not lead to any pellet formation upon centrifugation, thus supporting the formation of the ATS9R/ASO complex upon mixing the ATS9R and ASO. The NP was characterized with dynamic light scattering (DLS), transmission electron microscopy (TEM), and ultraviolet-visible (UV-Vis). DLS indicated that the hydrodynamic size of the NP was ~200 nm (Figure 3C, left). TEM images of the complex were collected with uranylacetate negative staining (Figure 3C, middle). The UV-vis profile of the complex indicated the presence of both the ATS9R and ASO in the complex with the corresponding peaks at 450 nm and 260 nm, respectively (Figure 3C, right). The ATS9R:ASO 1:1 NP complex was stable even after being kept in 4 °C for a month.

### 3.3. NP Functionally Reduced Chemerin Expression in Isolated Adipocytes

Using isolated adipocytes incubated with either 54 µg/mL ASO or 54 µg/mL ASO concentration within the NP, chemerin mRNA was significantly reduced in both Epi and RP fat adipocytes compared to the vehicle, regardless of treatment (Figure 4). The ASO concentration was determined by previous studies as an effective concentration to reduce chemerin expression in vitro [41]. The NP utilized a 1:1 formulation of ATS9R:ASO, creating an equal weight ratio of ATS9R to ASO within the NP complex. Our NP is therefore functional and can reduce chemerin mRNA in isolated adipocytes at a 1:1 ratio.

### 3.4. NP Reduced Chemerin Expression in Ex Vivo Incubated RP, but Not Epi Fat

The next set of experiments investigated whether this same NP could reduce chemerin expression in whole tissue. Adipose tissue samples were incubated with either 0.002 mg/mL ASO or 0.002 mg/mL 1:1 formulation of the NP (determined by ASO), using adipocyte maintenance media as a vehicle. This concentration was determined as a 10 mg/kg dose equivalent to allow for comparison to in vivo studies. The NP was able to significantly reduce chemerin mRNA in RP fat compared to the vehicle. The NP was not as effective at reducing chemerin in Epi fat (Figure 5), largely because of the variability within the vehicle Epi samples. Data in Appendix A supports that prohibitin, the hypothesized recipient of the ATS-labeled NP, is present in rat adipose tissues.

### 3.5. NP Was Unable to Selectively Deliver ASO to Fat In Vivo

We next moved to in vivo models to study the distribution of an injected NP as well as chemerin knockdown. Tissues were examined two days post-subcutaneous injection. This is a time point that is sufficient to observe reduced chemerin mRNA expression, as mediated by the ASO against chemerin [41]. Chemerin mRNA expression was significantly reduced in liver samples compared to the vehicle, both by the ASO alone and the 1:1 formulation of the NP at a 10 mg/kg dose (determined by ASO concentration). Reduction in chemerin was not observed in either Epi or RP fat when the 1:1 formulation of the NP was administered subcutaneously (Figure 6). There was a modest reduction of chemerin mRNA observed in fats when ASO alone was administered but this was not statistically significant. While the lack of significant reduction with ASO alone in adipose samples was surprising, this further emphasizes the importance of developing a drug delivery system that avoids the liver, due to most of the ASO functionality being shown in liver samples.

### 3.6. FITC Labeling Was Not a Viable Method of Tracking the NP

The FITC fluorophore attached to the ATS9R peptide of the NP had the promise of tracking the final destination of the NP when administered in vivo. Tissues were imaged on the IVIS after removal from rats given in vivo subcutaneous injections of the vehicle, the ASO, and 1:1 formulation of the NP at 10 mg/kg doses. This was done to measure the FITC radiance within each sample. No sample had significant FITC radiance over its vehicle or ASO alone treated counterparts (Figure 7A).

The lack of radiance was concerning, especially due to the NP’s ability to significantly reduce chemerin mRNA expression in the liver when given in vivo. To understand why this was the case, RP and Epi fat were incubated ex vivo with a 10 mg/kg equivalent concentration of NP and FITC-labeled ATS9R overnight and imaged the following day. The FITC signal was reduced in NP-treated tissues versus tissues incubated with FITC-labeled ATS9R alone (Figure 7B). This may be the result of an FITC quenching effect by the ASO and/or auto quenching by the FITC. The FITC fluorophore on the ATS9R peptide is thus not effective in tracing NP distribution.

### 3.7. Alternative NP Administration Route and Concentrations Did Not Improve Selectivity

Because the above in vivo studies used subcutaneous injection of NP and no adipose tissue specificity was observed, we next tested whether the injection route was preventing the NP from being more selectively bioavailable. The NP (1:1 ATS9R:ASO) was administered through a tail vein injection into male Dahl SS rats. Rather than improving specificity for adipose delivery, administration of the NP via IV injection showed similar results to NP subcutaneous injection. Specifically, liver but not Epi or RP fat chemerin mRNA expression was significantly reduced (Figure 8).

We next investigated if the concentration of ASO within the NP was too low in the 1:1 formulation to allow for adipose specificity. We also increased the dose of the NP to deliver a 25 mg/kg ASO, a dose that abolished circulating chemerin [27]. Reducing the ratio of ATS9R:ASO to 0.25:1, thus enhancing the amounts of ASO per particle, we administered both 10 and 25 mg/kg doses in a 0.25:1 formulation of the NP (determined by ASO concentration) subcutaneously. Two days post injection, we observed a dose-dependent liver chemerin mRNA reduction by the NP with a greater reduction when the 25 mg/kg dose was given compared to the 10 mg/kg dose. However, no fat specificity was observed as there was no significant reduction seen in either Epi or RP fat with either the 10 or 25 mg/kg dose (Figure 9).

## 4. Discussion

These experiments were conducted to determine whether an ATS9R drug delivery system could be used to reduce adipose tissue-derived chemerin selectively. Adipose tissue-derived chemerin is an important target because it is the primary source of genetic variations in *Rarres2* in the regulation of serum chemerin [42]. In the Dahl SS rat, the liver expressed the most chemerin mRNA, followed by adipose tissues. Within the adipose depots of Epi and RP fat, the adipocyte was the primary contributor of chemerin. We could consistently synthesize an NP comprised of a 1:1 ratio of ATS9R and ASO against chemerin and demonstrate that this NP was effective in reducing chemerin mRNA expression in isolated Epi and RP adipocytes in vitro. These findings provided a solid foundation for targeting the adipocyte for ASO delivery, and that the NP is chemically stable enough to do so.

### 4.1. Lack of Success of Adipose-Specific NP Delivery

However, this NP produced mixed results in an ex vivo setting. First, the NP in its 1:1 ATS9R:ASO ratio reduced chemerin mRNA expression in RP, but not Epi fat. Lack of reduction in Epi fat is likely because of the variability in measures of the vehicle group. Second, this NP was unable to deliver the ASO to adipose tissue selectively over the liver when administered in vivo subcutaneously or through tail vein injection. The NP was robustly taken up by the liver, a tissue infamous for its reticular endothelial system that serves the function of clearance. Our approach directly followed that used in mice [30,32]. However, we were unable to replicate the adipose selectivity of NP development in the rat. While both species express prohibitin on adipocytes [31] (Appendix A), there may be other species differences that interfere with NP adipose tissue specificity within the rat. It would be ideal to construct an ASO that had a tag for an adipocyte-specific protein, just as the GalNAc tag functions for the asialoglycoprotein receptor on hepatocytes. However, to our knowledge, no such tag exists. This was the driving force behind these studies adopting the use of the ATS9R NP containing the ASO. Avoiding the liver in attempts to reduce chemerin in particular is important with recent discoveries, such as those made by Stelmanska [43]. This study in rats determined that chemerin gene expression was regulated by nutritional status in the rat adipose tissue, not in the liver. This reaffirms the idea that biologically active chemerin is likely not made by the liver, but by other tissues that include adipose tissue.

### 4.2. Modifications of NP Administration and Formulation Did Not Improve Adipose-Specific Delivery

Efforts to track the NP via the FITC tag on the ATS9R peptide were difficult because of the quenching the ASO appears to have on FITC radiance. Additionally, changing the in vivo NP administration route to the tail vein IV did not improve adipose tissue specificity, nor did increasing the relative concentration of ASO within the NP to a ratio of 0.25:1 ATS9R to ASO, both at 10 and 25 mg/kg doses. We must conclude that the ATS9R drug delivery system is not specific for adipose tissue drug delivery in rats.

### 4.3. Challenges, Limitations, and Alternatives

We are cognizant of the struggle in the NP field of avoiding NP uptake by the liver. Future directions to improve adipose specificity over the liver can include masking the liver prior to NP administration [44]. Described as a “don’t eat me system”, this could allow for reduced uptake of the NP by the liver, subsequently making more of the NP available for adipose tissues. Other formulations of the NP, such as further increasing the ASO concentration within, or increasing the number of NP injections may allow for greater specificity of the NP to the fat vs. the liver as well. It is unclear whether differences in the reticular endothelial system of the mouse and rat are sufficiently different so as to explain the observed outcomes. We continued in this study, even given our negative findings, because it was our intent to demonstrate, in a deliberate and methodical way, that this published means of adipose-specific delivery cannot be reproduced by our hands. This, in our opinion, is essential to communicate in helping scientists choose differently when considering adipose targeted delivery, at least in the rat. Along these lines, a recent finding by Liu et al. supports the expression of prohibitin at the plasma membrane of human hepatoma cells [45]. This raises the possibility that the ATS sequence could drive the NP to be taken up by the hepatocyte. Such a possibility makes it difficult to understand how an ATS NP has been used selectively in the mouse. To achieve our intended goal of knocking down adipose-derived chemerin selectively, an adenovirus targeted to adipose tissue chemerin in vivo may be necessary [46].

We note several limitations of the present studies and alternatives to consider. All ex vivo studies described in this paper were only accomplished on adipose tissue samples. This is due to the inability of liver tissue to remain viable in an ex vivo setting, independent of the media in which the livers are incubated. We recognize and appreciate that there was variability within our vehicle samples that led to us not being able to conclude whether the NP could effectively knockdown the chemerin ex vivo (Figure 5). Use of a small number of rats did not, however, preclude us drawing a conclusion that the NP could effectively knockdown chemerin in the liver (but not fats). An alternative approach includes giving additional doses of the NP to increase the potential for the fats to take up the NP, but this would not obviate the outcomes that refute the hypothesis that this NP can deliver the ASO specifically to the fats. These studies were also exclusively conducted in male rats. Future studies might focus on including female animals as well, but their reduced fat burden may provide an additional challenge in achieving adipose specificity of NP delivery. Both sexes were not necessary in what was considered technical development and this reduced the number of animals we used. However, the data support a sex difference in the amount of chemerin protein measured in the plasma, with males having higher levels than females [47]. It is unclear how, in the rat, secretion of chemerin from the liver vs. the adipocyte differs. In the mouse, chemerin from the liver appears to be constitutively secreted. By contrast, chemerin in the adipose tissue, but not the liver, can be stimulated by inflammogens such as TNF alpha [22]. This implicates adipose-specific production of chemerin in obesity, a disease well recognized for hallmarks of inflammation. This also supports that chemerin regulation in the liver and adipose tissues is not the same. If these differences could be understood, interventions that modify adipose but not liver-derived chemerin could be beneficial tools in studying adipose contributions of chemerin to cardiovascular disease. Finally, an adipose-specific chemerin mouse knockout has been made [48]. Interestingly, the loss of adipocyte chemerin improved glycolipid metabolism in these mice on a high fat diet. Similar tools in the rat (e.g., floxed rats) are slowly being developed and could be turned to if a system to deliver the ASO against chemerin specifically to the adipocyte cannot be made.

### 4.4. Summary

This study, depicted pictorially in Figure 10, has important implications. First, it emphasizes the importance of adipose tissue, and specifically the adipocyte, as a potential source of biologically active chemerin that contributes to blood pressure regulation. Second, this work describes the difficulties of developing a drug delivery system that is adipose tissue-specific using the ATS9R peptide in vivo in the rat. We could not reproduce the adipose selectivity achieved in the mouse. As such, other means to deliver substances specifically to the adipocyte in the rat must be developed/considered.

## Figures and Tables

**Figure 1 biomedicines-10-01635-f001:**
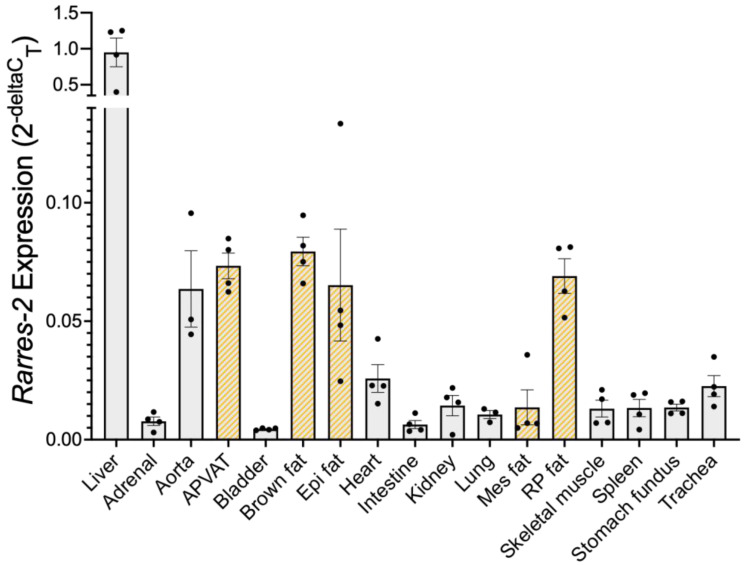
Chemerin mRNA expression within naïve male Dahl SS tissues. Gold diagonal stripes indicate adipose tissue samples. *β-actin* was used as a reference gene. Data are expressed as mean ± SEM. *n* = 3–4 rats for each group.

**Figure 2 biomedicines-10-01635-f002:**
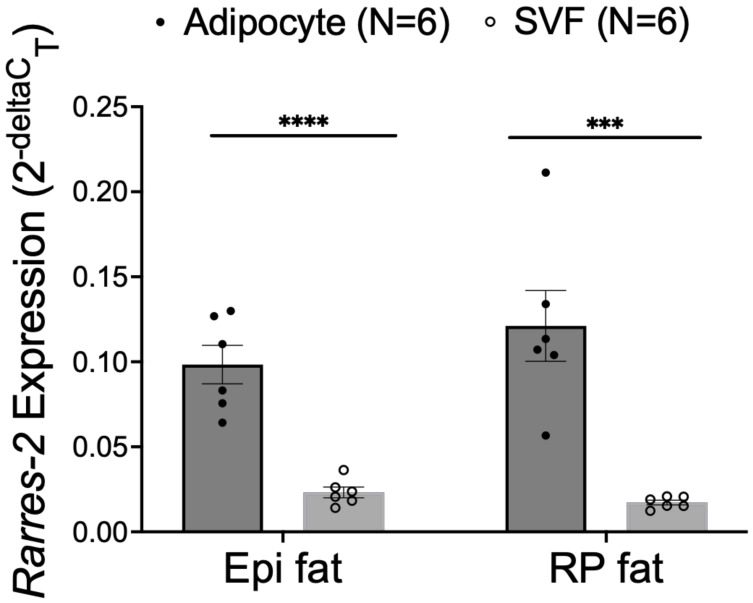
Chemerin mRNA expression in naïve Dahl SS adipocytes and stromal vascular fraction (SVF) isolated from Epi fat and RP fat. *β-actin* was used as a reference gene. Data are expressed as mean ± SEM. *n* = 6 rats for each group. **** *p* < 0.0001 compared to SVF samples and *** *p* < 0.001 compared to SVF samples determined by unpaired *t*-test.

**Figure 3 biomedicines-10-01635-f003:**
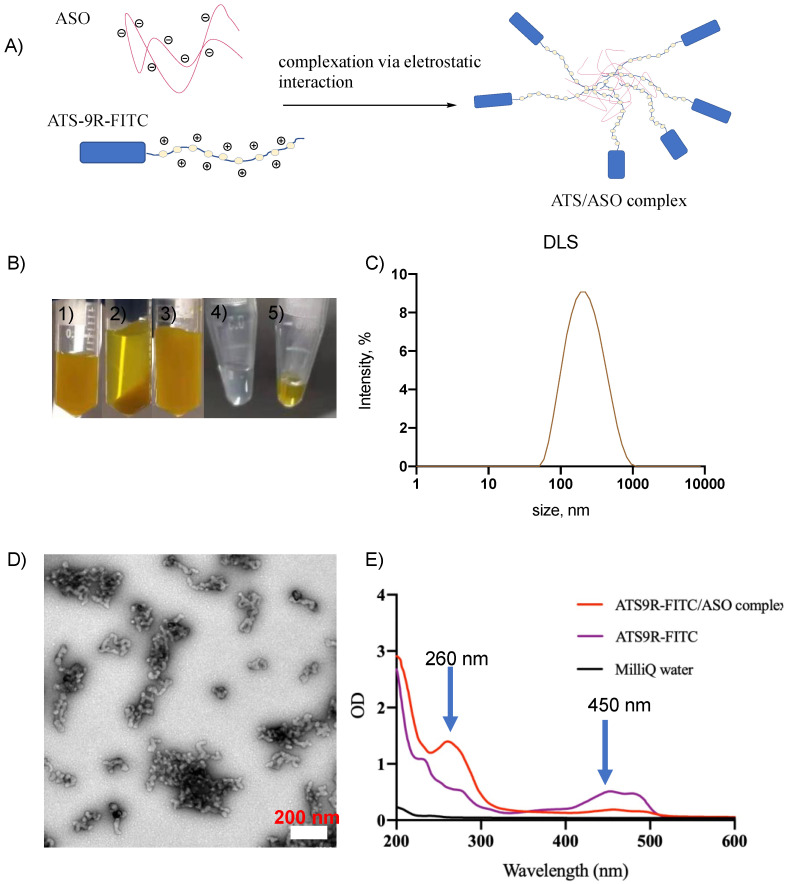
Synthesis of the characterization of the ATS9R/ASO complex. (**A**) The synthesis was achieved through the electrostatic interaction with ATS9R and ASO by mixing ATS9R and ASO under vigorous stirring. (**B**) Images of the ATS9R/ASO complex at processing stages: (1) ATS9R/ASO solution after complexation; (2) ATS9R/ASO solution after centrifugation; (3) pellet was collected and redispersed back into water; (4) 10 mg/mL ASO solution after centrifugation. No pellet formation was observed; (5) 10 mg/mL ATS9R solution after centrifugation. No pellet formation was observed. These results support the successful formation of the NP through ATS9R/ASO complexation. (**C**) The NP complex was characterized with DLS, TEM, and UV-vis. DLS indicates the hydrodynamic size to be around 200 nm ((**C**), left). TEM images were obtained with uranyl acetate staining ((**D**), scale bar 200 nm). UV-vis quantification was carried out ((**E**), right) with arrows indicating peaks at 260 nm and 450 nm for ASO/FITC-labeled peptide, respectively. Based on the individual standard curves of ASO- and FITC-labeled peptide, the amounts of the ASO and peptides were calculated. The relative molar ratio of the ASO/peptide was determined to be 1:1. MilliQ water serves as a vehicle control.

**Figure 4 biomedicines-10-01635-f004:**
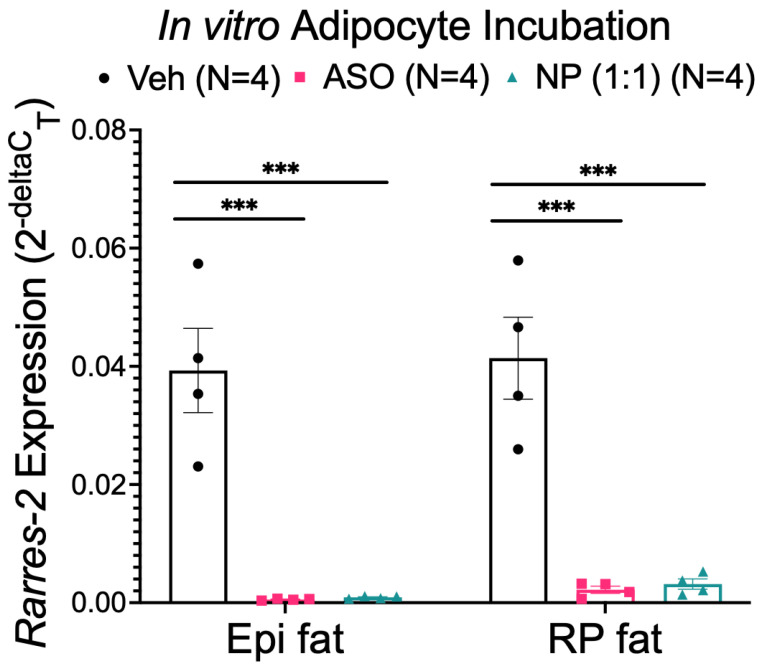
Chemerin mRNA expression in Epi and RP fat isolated adipocytes (from male Dahl SS rats) incubated with the vehicle, ASO, or NP overnight. *β-actin* was used as a reference gene. Data are expressed as mean ± SEM. *n* = 4 rats for each group. *** *p* < 0.001 compared to the vehicle treated samples by one-way ANOVA with Tukey’s multiple comparison.

**Figure 5 biomedicines-10-01635-f005:**
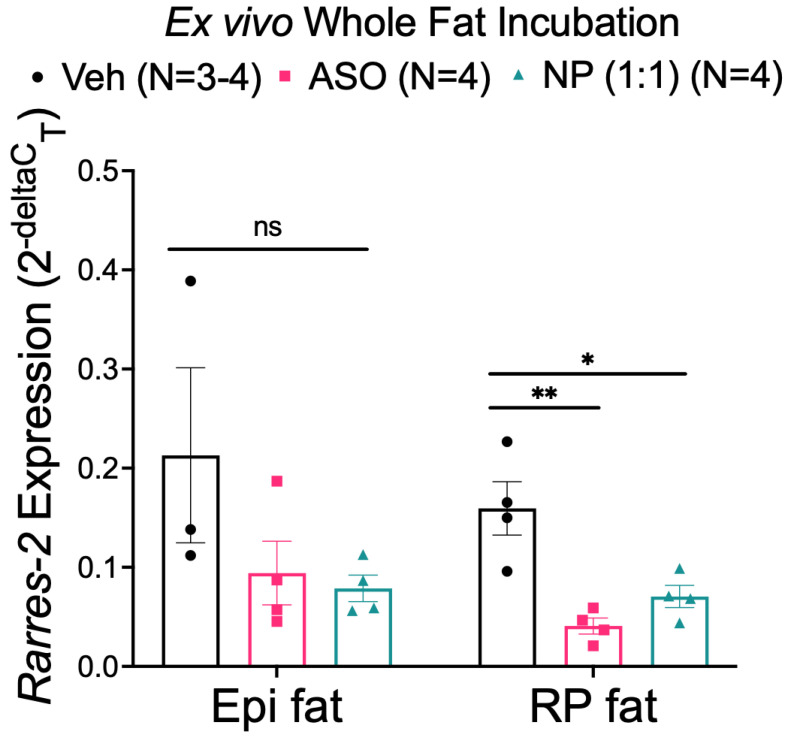
Chemerin mRNA expression in Epi and RP fat (from male Dahl SS rats) incubated with the vehicle, ASO, or NP overnight. *β-actin* was used as a reference gene. Data are expressed as mean ± SEM. *n* = 3–4 rats for each group. * *p* < 0.05 compared to the vehicle-treated samples and ** *p* < 0.01 compared to the vehicle-treated samples by one-way ANOVA with Tukey’s multiple comparison.

**Figure 6 biomedicines-10-01635-f006:**
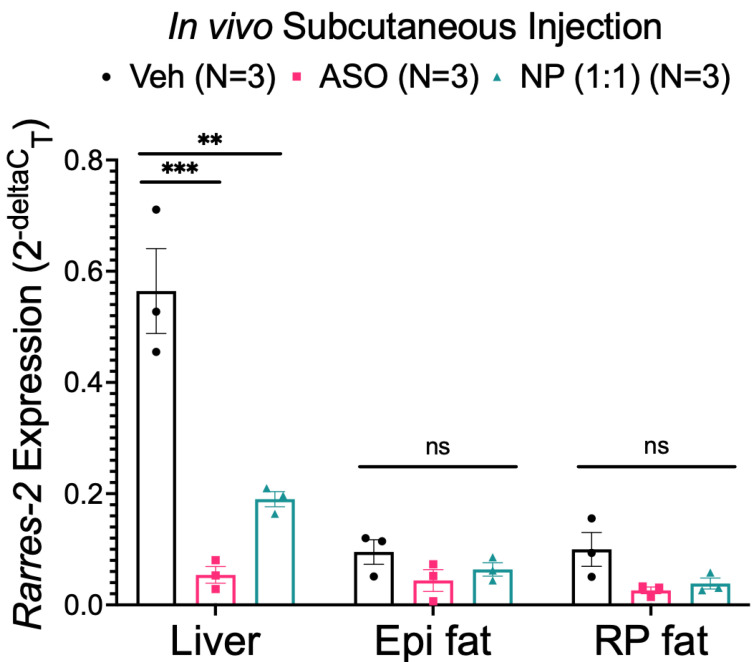
Chemerin mRNA expression in liver, Epi, and RP fat samples (from male Dahl SS rats) after in vivo subcutaneous injection with the vehicle, 10 mg/kg ASO, or 10 mg/kg NP with 1:1 ratio of ATS9R:ASO. *β-actin* was used as a reference gene. Data are expressed as mean ± SEM. *n* = 3 rats for each group. *** *p* < 0.001 compared to the vehicle-treated samples and ** *p* < 0.01 compared to ASO-treated samples by one-way ANOVA with Tukey’s multiple comparison.

**Figure 7 biomedicines-10-01635-f007:**
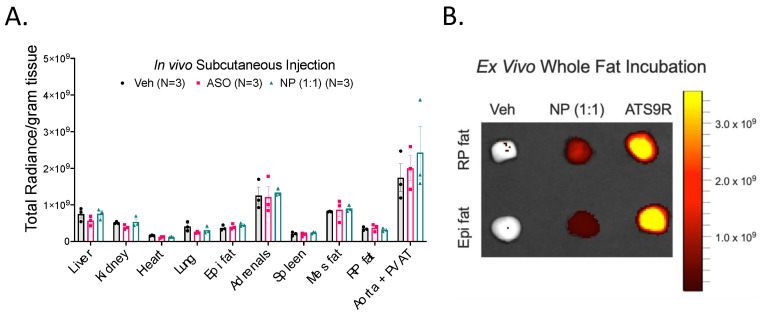
Radiance measures of FITC fluorescence in tissues treated with the vehicle, ASO, or NP with a 1:1 ratio of ATS9R:ASO in vivo via subcutaneous injection, normalized to grams of tissue weight (**A**). Data are expressed as mean ± SEM. *n* = 3 rats for each group. (**B**) Image of FITC radiance within RP and Epi fat samples incubated ex vivo with the vehicle, 1:1 NP, or ATS9R overnight.

**Figure 8 biomedicines-10-01635-f008:**
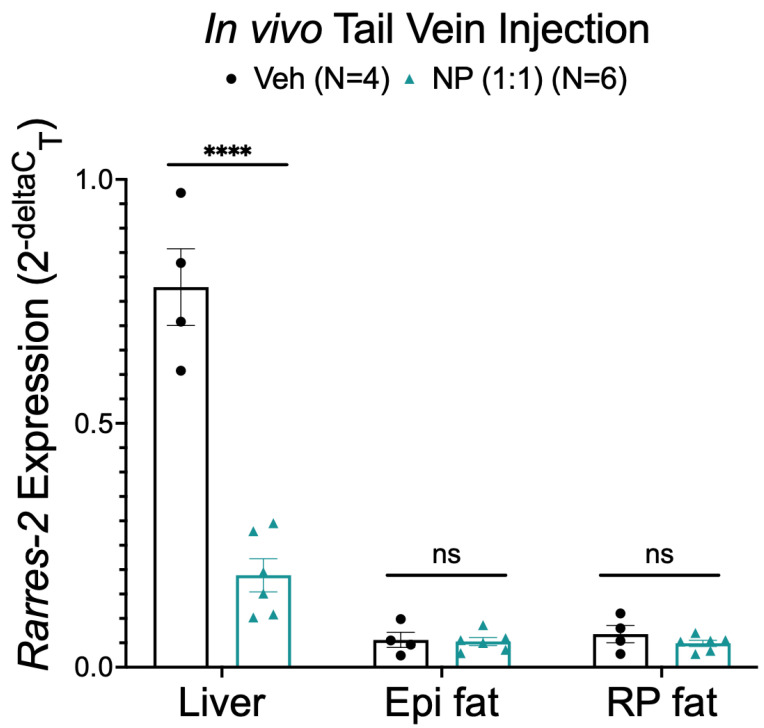
Chemerin mRNA expression in liver, Epi, and RP fat samples (from male Dahl SS rats) after in vivo treatment with the vehicle, 10 mg/kg ASO, or 10 mg/kg NP with a 1:1 ratio of ATS9R:ASO administered through tail vein IV injection. *β-actin* was used as a reference gene. Data are expressed as mean ± SEM. *n* = 4 rats for the vehicle-treated group and *n* = 6 for the NP-treated group. **** *p* < 0.0001 compared to the vehicle-treated samples by one-way ANOVA with Tukey’s multiple comparison.

**Figure 9 biomedicines-10-01635-f009:**
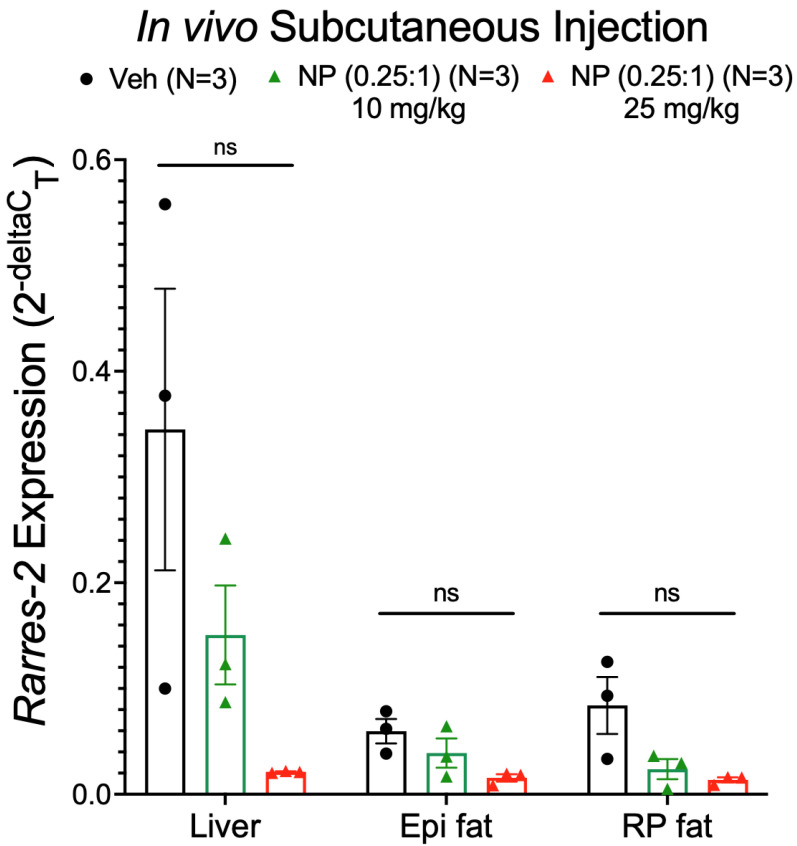
Chemerin mRNA expression in liver, Epi, and RP fat samples (from male Dahl SS rats) after in vivo subcutaneous injection with 10 mg/kg NP with a 0.25:1 ratio of ATS9R:ASO, 25 mg/kg NP with a 0.25:1 ratio of ATS9R:ASO, or the vehicle. *β-actin* was used as a reference gene. Data are expressed as mean ± SEM. *n* = 3 rats for each group. No significant differences between samples were detected by one-way ANOVA with Tukey’s multiple comparison.

**Figure 10 biomedicines-10-01635-f010:**
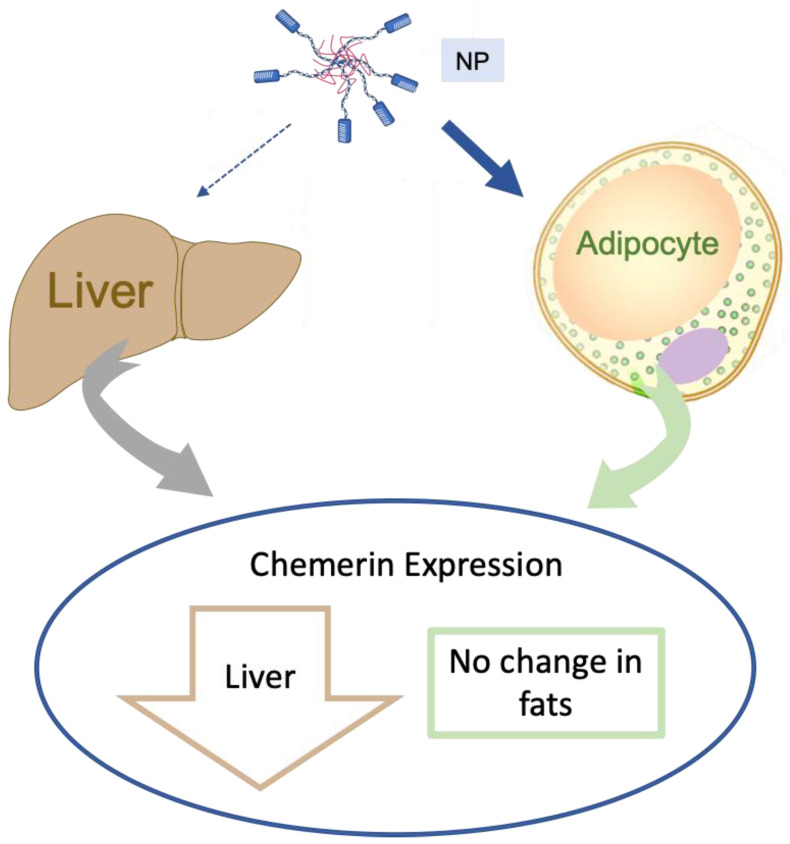
Depiction of the inability to knockdown adipose tissue-derived chemerin with avoidance of liver chemerin knockdown by the ATS9R NP carrying the ASO against chemerin. Width/intensity of blue arrows depicts the intent of this study, while the blue circle depicts our experimental findings.

## Data Availability

Data are available upon request.

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
