# Peer review of "Divergence of Chemerin Reduction by an ATS9R Nanoparticle Targeting Adipose Tissue In Vitro vs. In Vivo in the Rat"

_biomedicines, 2022, doi:10.3390/biomedicines10071635_

Round 1

Reviewer 1 Report

I thank the authors for the reply to all my questions and appreciate the honest and open answers. Additionally, the modifications clarified some parts of the manuscript.

Regarding the different ex vivo results between epiFat and RpFat authors stated that the best explanation is the higher variability of the Vehicle for the Epi fat, which I agree. However, since a group of only 3 rats were used in these experiments, I believe that it would be better to increase the n and be sure of the results instead of concluding that the NP was not as effective in reducing chemerin expression in Epi fat as in Rp fat.

Additionally, understanding why the liver is the main target of the NP in vivo, which is strictly linked to the specificity of the NP targeting, should be clarified.

Now, it is clearer to me that the major aim of the work was to highlight the difference of mice and rats in the uptake of the NP. Although I understand that this information is useful I think that the impact of the manuscript is below the threshold required for publication in Biomedicines.

Author Response

Response to Reviewer #1

biomedicines-1776813R1

Divergence of Chemerin Reduction Abilities of ATS9R Nano-particle Targeting Adipose Tissue in vitro vs in vivo in the Rat.

We have responded to your criticisms.  Your comments are in bold, while ours are in normal font.  You will find changes in RED in the resubmitted revision.

  • Regarding the different ex vivo results between epiFat and RpFat authors stated that the best explanation is the higher variability of the Vehicle for the Epi fat, which I agree. However, since a group of only 3 rats were used in these experiments, I believe that it would be better to increase the n and be sure of the results instead of concluding that the NP was not as effective in reducing chemerin expression in Epi fat as in Rp fat.

We are unable to do more experiments in the time that we’ve been given to submit this revision.  Additionally, the outcome of these additional experiments would not modify the primary outcome of the study, namely that this NP – designed to be adipocyte specific- was not adipocyte specific.  We spoke with our attending veterinarian in the honor of reduce, replace and refine, and the consensus was it would be difficult to justify sacrificing more animals for an experiment that would not change the outcome.  As such, we have not increased the N for these experiment. We hope this acceptable to you. 

  • Additionally, understanding why the liver is the main target of the NP in vivo, which is strictly linked to the specificity of the NP targeting, should be clarified.

We are not quite clear on what you mean, but believe its wishing for more of an explanation as to why the NP would be taken up by the liver?  We have done our best in the revision to do so within the first paragraph of the discussion. 

  • Now, it is clearer to me that the major aim of the work was to highlight the difference of mice and rats in the uptake of the NP. Although I understand that this information is useful I think that the impact of the manuscript is below the threshold required for publication in Biomedicines.

With all respect, you misunderstand.  This was NOT the primary purpose of this study but rather something we discovered.  If our purpose was to highlight the difference in mice and rats, we would have done this study with both mice and rats.  We did not, as the primary purpose was to determine if the ATS9R NP could be specifically target to the fat in the rat.  We have revised several places within the text to make this more clear.  We agree with you that the way these sentences were written could convey the idea that the purpose of the manuscript was to highlight differences between mice and rats.  To be clear, our study DOES support the idea that maybe the mouse and rat are different…but the most important point is that we could not reproduce one of the sole ways a NP has been used to successfully deliver a substance to adipose tissue. 

Reviewer 2 Report

No further comments.

Author Response

Response to Reviewer #2

biomedicines-1776813R1

Divergence of Chemerin Reduction Abilities of ATS9R Nano-particle Targeting Adipose Tissue in vitro vs in vivo in the Rat.

Thank you for your acceptance of our responses to you.

Reviewer 3 Report

The authors have systematically investigated FITC-labelled ATS9R Nanoparticles carrying an antisense oligonucleotide (ASO) against chemerin to reduce chemerin expression and study obesity-associated hypertension. The authors have carelessly submitted files and not the final format, however, the work reported in this manuscript is interesting and well presented. There are many grammatical and sentence errors in the article, and the language organization needs to be improved. For these reasons, I conclude that the paper should undergo minor revisions.

1.      Verify the files properly before submitting them.

2.      Rewrite the sentence “Using in vitro, ex vivo, and in vivo techniques, we tested whether a FITC-labelled ATS9R NP carrying an ASO against chemerin would reduce chemerin expression in a tissue selective manner, being more bioavailable to adipose vs liver tissues.” the sentence has structurally ambiguous grammar.

3.      In Figure 3, TEM image resolution is low with no scale bar. Provide the figure with a higher resolution with a scale bar.

4.      UV spectrophotometry of ATS9R NP without ASO have be provide as a control sample.  

5.      The manuscript contains some typographical errors and superfluous spaces that need to be corrected accordingly.

Author Response

Response to Reviewer #2

biomedicines-1776813R1

Divergence of Chemerin Reduction Abilities of ATS9R Nano-particle Targeting Adipose Tissue in vitro vs in vivo in the Rat.

The authors have systematically investigated FITC-labelled ATS9R Nanoparticles carrying an antisense oligonucleotide (ASO) against chemerin to reduce chemerin expression and study obesity-associated hypertension. The authors have carelessly submitted files and not the final format, however, the work reported in this manuscript is interesting and well presented. There are many grammatical and sentence errors in the article, and the language organization needs to be improved. For these reasons, I conclude that the paper should undergo minor revisions.

  1. Verify the files properly before submitting them.

We submitted this manuscript precisely how Biomedicines required and we take incredible, purposeful care when submitting any manuscript.  We are required to submit a Word file that is not locked.  This means that you can see the review/revision side of things if opened in Review and this, clearly, would not look like the final form unless you go to “No Markup”.  All we can surmise is that you opened the paper in this view, vs the view of no comments.  The other two reviewers saw our paper as a final version, and were able to comment on it appropriately.  Moreover, we did not hear from Biomedicines that our files were not appropriate. 

  1. Rewrite the sentence “Using in vitro, ex vivo, and in vivo techniques, we tested whether a FITC-labelled ATS9R NP carrying an ASO against chemerin would reduce chemerin expression in a tissue selective manner, being more bioavailable to adipose vs liver tissues.” the sentence has structurally ambiguous grammar.

Done as requested; stated below.

We tested whether a FITC-labelled ATS9R NP carrying an ASO against chemerin would reduce chemerin expression in an adipose selective manner.  This ATS9R NP carrying the ASO was used in complimentary in vitro, ex vivo, and in vivo experiments. 

  1. In Figure 3, TEM image resolution is low with no scale bar. Provide the figure with a higher resolution with a scale bar.

The previous TEM image did have a scale bar but it was too small. We have expanded the size of the scale bar. With the enlarged TEM image, the resolution is better.  Thank you for noticing this.

  1. 4.      UV spectrophotometry of ATS9R NP without ASO have be provide as a control sample.  

UV-vis spectrum of ATS9R-FITC has been added to the figure. For ATS9R-FITC/ASO complex, the absorbance maximum at 260 nm mostly comes from ASO component, while the absorbance maximum at 450 nm mostly comes from ATS9R-FITC.

.

  1. The manuscript contains some typographical errors and superfluous spaces that need to be corrected accordingly.

We have read this manuscript again carefully and done our best to correct errors.  The spaces have constantly been a problem because of the template with which we are required to work for this journal.  We have done our best. 

Round 2

Reviewer 1 Report

I thank the authors for the reply.

Author Response

Much appreciated that you took the time to do this with and for us.  We are honored that you did so.

This manuscript is a resubmission of an earlier submission. The following is a list of the peer review reports and author responses from that submission.

Round 1

Reviewer 1 Report

In the present study, Orr A and colleagues tested a new delivery method, specific to adipose tissue, of an antisense oligonucleotide (ASO) directed to chemerin. Authors used a nanoparticle (NP) containing a targeting sequence to prohibitin, an adipocyte cell surface protein, to specify the delivery to adipose tissue. The aim was to reduce the levels of chemerin specifically secreted by adipose tissue. Although the in vitro and ex vivo analysis were promising, in vivo data was not successful. Overall, this is an interesting subject with a clinical relevance and application, since the adipocyte-derived chemerin contributes to blood pressure regulation. The manuscript is clearly written and results are carefully presented. However, some conclusions regarding the lack of success of the in vivo studies taken by the authors are not fully supported by the experimental data. Additionally, the relevance of this method is questionable since an adipo-specific chemerin mouse knockout was produced, as indicated by the authors.

Comments:

  • A major question regards the preferential in vivo delivery of the NP to liver. The adipocyte specificity of NP targeting relies on the presence of the targeting sequence to prohibitin, a protein located at the adipocyte cell surface. However there is data indicating that this protein is also expressed in the liver, at the cell membrane (http://dx.doi.org/10.1016/j.ebiom.2015.09.018). Considering this how do authors ensure the specificity of the NP targeting to adipose tissue?
  • The content of the NP, the ATS9R/ASO complex, needs a deeper characterization. Although the same concentration of ATS9R and ASO solutions were used in order to establish the complexes, the concentration of the individual components in the NP (the ASO and the peptide) were not measured and it is not possible to indicate that they establish a 1:1 ratio complex.
  • An immunoprecipitation experiment could be performed in order to conclude if ASO and ATS9R-FITC form a complex
  • How do authors explain the different ex vivo results between epiFat and RpFat (figure 5?)

Reviewer 2 Report

The manuscript submitted by Alexis Orr et al. aimed to identify a novel approach to knock down adipose tissue-derived chemerin without interfering with the liver-derived expression of Chemerin using the nanoparticle (NP) approach. Using this novel NP-based approach, the authors showed that the knockdown was efficiently achieved for the liver; however, adipose-specific knockdown invivo was unsuccessful. However, in isolated adipocytes, the knockdown was achieved. Overall, the manuscript is well written and emphasizes the challenges in drug delivery research in this context, its Chemerin, and its physiological relevance to fat metabolism – hypertension. 

Minor comments: 

  1. It looks like chemerin expression and secretion from liver vs. adipocytes differ entirely; how the circulating levels of chemerin are regulated? What are the potential sites of the target?
  2. Is there any sex-dependent effect on Chemerin expression that was studied/reported?
  3. Is there any report published on chemerin in mice in the event of HFD-induced metabolic abnormality followed by high blood pressure?
  4. Did the authors try adenovirus-specific delivery to target adipose tissue-specific chemerin in invivo?